# Cancer and Potential Prevention with Lifestyle among Career Firefighters: A Narrative Review

**DOI:** 10.3390/cancers15092442

**Published:** 2023-04-24

**Authors:** Amalia Sidossis, Fan-Yun Lan, Maria S. Hershey, Kishor Hadkhale, Stefanos N. Kales

**Affiliations:** 1Department of Environmental Health, Harvard T.H. Chan School of Public Health, Boston, MA 02138, USA; amaliasidossis@gmail.com (A.S.); flan@hsph.harvard.edu (F.-Y.L.); mhershey@hsph.harvard.edu (M.S.H.); khadkhale@hsph.harvard.edu (K.H.); 2Division of Occupational Medicine, Cambridge Health Alliance/Harvard Medical School, Cambridge, MA 02139, USA; 3Institute of Health and Welfare Policy, National Yang Ming Chiao Tung University, Taipei 112, Taiwan; 4Occupational Medicine, Department of Family Medicine, National Yang Ming Chiao Tung University Hospital, Yilan 260, Taiwan; 5Faculty of Social Sciences Health Sciences (Epidemiology), Tampere University, 33100 Tampere, Finland

**Keywords:** cancer, firefighters, incidence, mortality, prevention, perception, lifestyle

## Abstract

**Simple Summary:**

Firefighting has been identified as a high-risk occupation with increased risks for several types of cancers. Exposure to carcinogens in fire smoke and within the fire station, occupational factors such as shift work, and lifestyle are all plausible contributors to disease development. Prevention strategies and a change to fire service culture are crucial for primary cancer prevention among career firefighters.

**Abstract:**

Career firefighters are at considerable risk for chronic diseases, including an increased risk of various cancers, compared to the general population. Over the last two decades, several systematic reviews and large cohort studies have demonstrated that firefighters have statistically significant increases in overall and site-specific cancer incidence and site-specific cancer mortality compared to the general population. Exposure assessment and other studies have documented exposures to a variety of carcinogens in fire smoke and within the fire station. Other occupational factors such as shift work, sedentary behavior, and the fire service food culture may also contribute to this working population’s increased cancer risk. Furthermore, obesity and other lifestyle behaviors such as tobacco use, excessive alcohol consumption, poor diet, inadequate physical activity, and short sleep duration have also been associated with an increased risk of certain firefighting-associated cancers. Putative prevention strategies are proposed based on suspected occupational and lifestyle risk factors.

## 1. Introduction

Firefighting is a high-risk occupation with many potential adverse health effects, including the development of various cancers. Several studies have thoroughly evaluated cancer incidence and mortality among firefighters across the globe, but the results have not always been consistent. When comparing career firefighters to the general population or other occupational groups, most authors report an excess in overall cancer incidence [1,2,3,4]. This excess risk is most evident when considering specific cancer sites; meanwhile, an overall reduction in cancer mortality may be explained by the “healthy worker effect” [1], which is a specific type of selection bias where the healthier individuals are more likely to be employed than their less healthy counterparts as a part of the recruitment and screening process, while an individual too sick to work has a higher probability of not being selected or resigning earlier. Accordingly, a working group of 25 international experts convened by the International Agency for Research on Cancer (IARC) classified occupational exposure as a firefighter as carcinogenic to humans (Group 1) [2]. The group also reported strong mechanistic evidence of genotoxic effects such as epigenetic alterations, oxidative stress, chronic inflammation, and modulating receptor-mediator effects being induced or presenting in this occupation [2]. Existing evidence suggests that several factors may contribute to disease development. Occupational exposures, such as diesel exhaust, polycyclic aromatic hydrocarbons (PAHs), perfluoroalkyl substances (PFAS), polybrominated diphenyl ethers (PBDEs), and formaldehyde, among others, are associated with harmful outcomes. In addition, poor lifestyle habits including poor nutrition, sedentary behavior, shift work, alcohol consumption, inadequate education, inadequate utilization of available healthcare facilities, and improper decontamination practices all serve as potential contributors to disease formation [3,4,5,6,7]. Most recent reviews (Appendix A) focus on typical occupational exposures related to firefighting but fewer emphasize the role of unhealthy cultures and behaviors among career firefighters. This review aims to provide a comprehensive overview of the available evidence on cancer incidence and mortality among career firefighters and to investigate the potential causes. In addition, we discuss cancer prevention methods and firefighters’ perceptions of the risks, while emphasizing the importance of lifestyle modification as a putative prevention strategy.

## 2. Methods

The present narrative review summarizes evidence from original studies and reviews on career firefighters published during the past ten years. When evaluating firefighter cancer incidence, comparing studies and consolidating evidence can sometimes be challenging due to heterogeneity in research design and subject characteristics, such as exposure type, employment duration, study location, and types of cancer examined. Therefore, only research on career firefighters was included in this review. Additionally, regarding the heterogeneity in firefighters’ working conditions across the globe, we categorized the literature found based on their study site by continent (i.e., North America, Europe, Oceania, and Asia). For the purpose of validity, studies from the different continents North America, Europe, Oceania, and Asia are considered separately. Studies were selected based on firefighters’ various carcinogenic exposure at work and the outcome of cancers. Additionally, the studies may have also included other risk factors such as lifestyle-related, shift work, and sleep quality as related to firefighters’ health culture. Our search strategy comprised primarily the reviews and original publications in PubMed from the past ten years. Additionally, 9 articles were included based on their importance. We retrieved a total of 177 articles and included 84 in this review. Inclusion and exclusion criteria are described in the flow diagram (Figure 1).

## 3. Results

### 3.1. Cancer Incidence

Most studies have reported an excess in overall cancer risk [8,9,10] compared to the general population or other occupations, while a few suggested a decreased risk of overall cancer [11,12]. When examining specific types of cancer, the evidence of increased incidence is more consistent.

### 3.2. Overall Cancer Incidence

We found that ten studies (Table 1) followed distinct career firefighter cohorts and reported overall cancer incidence risk [1,8,9,10,11,13,14,15,16,17]: one conducted in North America [18], five in Europe [8,11,14,15,16], three in Oceania [1,10,17], and one in Asia [13]. American career firefighters have an almost 10% excess overall cancer incidence for males and 24% excess for females when compared with the US general population [9], which agrees with most studies in Europe (2–6% excess overall cancer incidence) [8,14,16] and Australia (9–15% excess in general; up to 85% excess for high-risk chronic exposure group) [1,10,17]. However, one Swedish [11], one Korean [12], and one Scottish study [15] observed the opposite, with a standardized incidence ratio (SIR 0.81, 95% CI 0.71–0.91) and 0.97 (95% CI 0.88–1.06), and an incidence rate of 86.5/10^5^/year (versus 123.7 for the general population, *p* < 0.001), respectively. Notably, most studies used the general population as the reference group; therefore, the “healthy worker effect” might have been introduced and caused an underestimation. To minimize the potential bias, the Danish study compared career firefighters to the working population or the Danish military and reported (SIR 1.02, 95% CI 0.96–1.09). Additionally, the role of long latency periods could play a significant role in the incidence of cancers.

### 3.3. Types of Cancer

Several studies have assessed the risk of both cancer incidence and mortality [19]. Results have not been entirely consistent, but generally indicate an increased risk of several cancer types among firefighters [19,20]. Specific cancer sites are described below.

#### 3.3.1. Skin Cancer

We found eleven studies that examined the rates of skin cancer among firefighters [1,8,10,11,14,15,16,17,21,22,23], of which ten specifically reported the risk of melanoma skin cancer [1,8,10,11,14,15,17,21,22,23]. All the studies were conducted in North America, Europe, or Oceania, and most of them found an increase in melanoma incidence among firefighters [8,10,14,15,17,21,22,23]; for example, Pukkala found a statistically significant excess case (SIR 1.62, 95% CI 1.14–2.23) in firefighters aged 30–49 years [8]. On the other hand, no excess risks were noted when looking into certain subgroups, such as firefighters older than 70 years old [8], and when using the military as the reference group [14]. It is likely that shared lifestyle factors between firefighters and military workers might be contributing risk factors for melanoma skin cancer development rather than firefighting-specific exposures (also evident in prostate and testicular cancer). In addition, Kullberg reported significantly lower risks for malignant skin melanoma among Swedish career firefighters (SIR 0.30, 95% CI 0.06–0.88), in contrast to most studies [11]. This may be explained by the selection of healthy individuals for the job and by the left truncation bias; enrollment to the cohort started in 1931 but the outcome could only be studied from 1958; therefore, all firefighters who died or emigrated between 1931 and 1958 were excluded from the cohort, leaving a slightly healthier population of firefighters at the start of follow-up. Additionally, unlike other Nordic countries, cancer incidences are incompletely registered in the Swedish cancer registry. This underreporting reduced by up to 4% in all incident cancer cases [24]. Regarding non-melanoma skin cancers, the current evidence is limited and mixed. Pukkala et al. found an increased risk, mainly in ages 70 years and older (SIR 1.40, 95% CI 1.10–1.76) [8]. Accordingly, Bigert et al. found a statistically significant excess of cases (SIR 1.48, 95% CI 1.20–1.80) [16], while no positive relationship between risk and work duration was noted [16], and Petersen et al. found no excess risk of non-melanoma skin cancers comparing Danish firefighters with the general population [14].

#### 3.3.2. Urogenital and Reproductive Cancers

Studies have reported cancers of the urogenital system among firefighters, including prostate, kidney, testicular, and female reproductive cancers (including breast cancer) [1,8,9,11,14,15,16,17,21,22,23,25,26]. Regarding prostate cancer, most studies were conducted in Oceania, North America, and Europe. Findings from these studies show an increased risk [10,16,21,22,23,25,26]. Despite the overall elevated risk of prostate cancer compared with the general population, a few firefighter subgroups have been shown to be at even greater risk, primarily in the age group of 30–49 years old [8], >70 years old [14], and non-Caucasian firefighters [9]. In particular, Pukkala et al. suggested a significantly increased incidence of prostate cancer (SIR 2.59, 95% CI 1.34–4.52) in ages 30–49 years, while no excess was found among older ages [8]. Daniels et al. reported an excess prostate cancer incidence (SIR 1.26, 95% CI1.02–1.54, *n* = 94) among non-Caucasian firefighters [9]. Petersen et al. noted a significant increase in prostate cancer in ages 70 years and older, while younger ages (<50 years) had fewer than expected cases [14]. Concerns about surveillance bias are notable for cancers such as prostate. If firefighters are screened frequently, they are more subject to detection and therefore demonstrate greater incidence rates [2]. Contrary to most evidence, a Swedish study suggested a lower risk for prostate cancer among firefighters (SIR 0.68, 95% CI 0.52–0.87) [11].

Bladder cancer is one of the most pronounced occupational cancer risks among firefighters. According to the IARC, there is sufficient evidence in humans (Group 1) for bladder risk among firefighters [2]. Findings from the epidemiological studies are consistent. In the UK-based study, Scottish male firefighters were observed with excess mortality of bladder cancer compared to the general population [27]. The Nordic study observed an 11% higher risk of bladder cancer compared to the general population [8]. However, the association was not statistically significant [28]. Similarly, a study from the USA observed a significantly increased risk of cancer incidence from 1985 to 2009 [9]. This study also observed a significantly increased risk of bladder cancer among female firefighters (SIR 12.5, 95% CI 3.41–32.05). However, the number of cases among female firefighters was very low in this study. Recent meta-analyses are in line with the findings from individual studies on bladder cancer risk in this occupational category [20,29,30].

Regarding kidney cancers, three studies conducted in Australia, Scotland, and the US were found [10,15,22]. Both the Scottish and US studies reported increased risks among firefighters when compared with the general population [15,22], with an approximately two-fold increased incidence, while no evidence of elevated kidney cancer risks was found in Australia. [10]. Likewise, a Canadian study reported statistically significant increased risk (HR 1.52, 95% CI 1.24–1.87) of kidney cancer among firefighters compared with other workers in the occupational disease surveillance system in Ontario [31].

Regarding testicular cancer, a few European studies found [8,14,15] mixed findings (SIR ranges from 0.51 to 1.23). Lastly, within limited evidence on female reproductive cancers (including breast cancer), Daniels et al. reported an excess breast cancer incidence among female firefighters, with an SIR of 1.45. Those aged 50–54 years were at the greatest risk [9].

#### 3.3.3. Other Cancers

There is some evidence of an increased risk of hematological cancers among firefighters [8,14,15,22,23], including multiple myeloma [8,22], acute myeloid leukemia [22], and Hodgkin’s lymphoma [14,23], based on research conducted in Europe and North America. On the other hand, the incidence of non-Hodgkin’s lymphoma did not increase when compared with the general population [14,15].

Respiratory cancer studies in Europe and North America provide mixed evidence. There is research showing increased lung cancer incidence among Caucasian male firefighters [9], adenocarcinoma of the lung [8], and mesothelioma [8], with reported SIRs, ranging from 1.1 to 2.6. However, there is also evidence showing a decreased lung cancer incidence overall [15] and among non-Caucasian male firefighters [9].

Finally, results on digestive cancers among firefighters were published in a few studies conducted in Europe and North America [9,11,14,15,16]. Elevated risks of stomach cancer [11,16] and esophageal cancer [9] have been reported, while the evidence on colorectal cancer is mixed and inconsistent [9,14,15].

### 3.4. Cancer Mortality

We found heterogeneity in the evidence of firefighters’ cancer mortality across different continents. In North American studies, elevated all cancer mortality was evident. Muegge et al. found that mortality due to all malignant cancers was 19% higher for Indiana firefighters compared to non-firefighters [32]. Likewise, Pinkerton et al. reported elevated all cancer mortality (SMR 1.12, 95% CI 1.08–1.16) among US firefighters from San Francisco, Chicago, and Philadelphia. The most pronounced risks observed were for mesothelioma (SMR 1.86, 95% CI 1.10–2.94), non-Hodgkin’s lymphoma (SMR 1.21, 95% CI 1.03–1.42), esophageal (SMR 1.31, 95% CI 1.10–1.55), intestine (SMR 1.27, 95% CI 1.14–1.40), rectum (SMR 1.32, 95% CI 1.07–1.61), lung (SMR 1.08, 95% CI 1.02–1.15), and kidney cancers (SMR,1.22, 95% CI 1.00–1.47), when compared with the general USA population [33] (Table 2).

On the other hand, evidence on Oceania, European, and Asian firefighters’ cancer mortality is mixed. Some studies suggested reduced overall cancer mortality or no difference when compared to the general population [10,11,21,34]. This might be the result of factors such as the “healthy worker effect” [1], or the fact that firefighters might have lower smoking rates compared to the general population. For example, firefighters have been shown to have lower smoking rates (<20%) than military personnel and other adult men [25]. The Nordic study observed that adjustments for lifestyle-related factors such as smoking and alcohol provided adjusted estimates which hold true risk estimates for many occupations including those related to firefighting (public safety workers) [35]. Regardless of the methodological limitations, SMRs across the three continents range from 0.58 to 1.12 (Table 2).

As to cancer mortality for specific cancer sites, there is evidence of increased mortality in firefighters’ urogenital and reproductive cancers (including kidney [9,13,15,32,33], bladder [9,13], and prostate cancer [36], gastrointestinal cancers (including stomach) [36] and colorectal cancer [13,33], respiratory cancers (including lung) [9,18,33], larynx and hypopharynx cancer [37], and mesothelioma [33], leukemia and lymphoma [13,18], and female firefighters’ breast cancer [9]. Notably, research has shown that increased cancer mortality is associated with firefighting duties. In particular, Daniels et al. suggested that firefighters’ lung cancer mortality risk increased with career fire runs (i.e., any response to a call that deployed the apparatus) (HR 1.11, 95% CI 0.95–1.29) and fire hours (the total time spent at fires) (HR 1.39, 95% CI 1.12–1.73) [18]. Accordingly, leukemia mortality risk increased with fire runs (HR 1.45, 95% CI 1.00–2.35) [18]. Lastly, there is little research evaluating cancer mortality among non-Caucasian male firefighters. Daniels et al. noted a decrease in total cancer mortality (SMR 0.80, 95% CI 0.65–0.97, *n* = 104) but an increase in prostate cancer mortality (SMR 1.64, 95% CI 0.95–2.63, *n* = 17) among the ethnic minority firefighter populations [9].

**Table 2 cancers-15-02442-t002:** Overall cancer mortality among firefighters.

Author/Year	Mortality Rate (95% CI)	Study Size	Study Location
North America			
Muegge 2018 [32]	OR = 1.19 (1.08–1.30)	2818 firefighters and 11,272 matched comparison death records	USA
Pinkerton 2020 [33]	MR = 1.12 (1.08–1.16)	29,992 career firefighters	USA
Europe			
Petersen 2018 [36]	SMR = 1.12 (1.00–1.26)	4659 full-time firefighters	Denmark
Amadeo 2015 [34]	SMR = 0.95 (0.88–1.02)	10,829 professional male firefighters	France
Ide 2014 [15]	Mortality rate/10^5^/year [SD] = 20.4 [27.4]	~2200 serving firefighters	Scotland
Zhao 2020 [37]	MRR = 1.00 (0.89–1.12)	27,365 firefighters	Spain
Oceania			
Glass 2016 [1]	SMR = 0.29 (0.01–1.64) (Low risk) SMR = 0.87 (0.40–1.65) (medium risk) ^a^ SMR = 1.47 (0.54–3.19) (High risk)	611 firefighters	Australia
Glass 2016 [10]	SMR = 0.81 (0.74 to 0.89)	17,394 full-time and 12,663 part-time male firefighters	Australia
Glass 2019 [17]	SMR = 0.83 (0.44–1.54)	1682 paid female firefighters	Australia
Asia			
Ahn 2015 [12]	SMR = 0.58 (0.50–0.68)	29,453 firefighters	Korea

MRR = mortality rate ratio, SMR = standardized mortality ratio, MR = mortality ratio, OR = odds ratio, CI = confidence interval, SD = standard deviation, HR = hazard ratio; ^a^ 154 out of 259 subjects in this category were volunteer firefighters.

### 3.5. Causes

Firefighters are exposed to various carcinogens during fires, within fire stations, and when improperly using and cleaning personal protective equipment. These exposures have been associated with adverse health effects including cancer, but the potential relationship with cancer development requires further research. Some of the most important occupational carcinogens encountered by firefighters are listed below.

#### 3.5.1. Diesel Exhaust

Diesel particulate matter is present in measurable concentrations within fire stations and specifically engine bays, duty offices, and dormitories; therefore, the fire station environment has the potential to contribute further to firefighter exposures to elemental carbon during fire ground activities [38]. It is very important that exposure is controlled and limited since diesel exhaust was classified as carcinogenic to humans (Group 1) by the International Agency for Research on Cancer in 2012 [39].

#### 3.5.2. Polycyclic Aromatic Hydrocarbons (PAHs)

Evidence has demonstrated that occupational exposure to PAHs has been associated with various cancers, due to their mutagenic and genotoxic properties [40]. It has been found that the main sources of PAH exposure in non-fire situations are fuel and biomass combustion, vehicular traffic emissions, and lubricant oils [40]. Oliveira and colleagues measured the levels of 16 PAHs in the breathing zones of workers at five Portuguese fire stations during a normal shift and found that the levels exceeded the WHO health-based guideline level, emphasizing that this exposure should be carefully monitored [40]. It appears that the main PAH exposure route is via skin absorption [41]. Stec et al. collected wipe samples from skin, firefighters’ personal protective equipment (PPE), and the work environment. Most of the studies on PAHs showed significant increases in concentration in all sampled locations, especially the hands. The authors concluded that the significantly increased levels of PAHs found on the samples taken from masks show insufficient cleaning procedures and therefore stringent protocols are needed to control those exposures [41]. Regarding particular PAH chemicals detected, Baxter et al. measured PAH levels during overhaul events in firehouses, using a University of Cincinnati administrative facility as a comparison location, and reported that among the 17 PAHs analyzed, only naphthalene and acenaphthylene were generally detectable, and specifically naphthalene was present in seven out of eight overhaul activities, in two out of four firehouse samples, and in none of the samples collected from the control site [42]. A greater number of PAHs were found in face and neck wipe samples, some of which have a carcinogenic activity such as benzo fluoranthene, which was also found in overhaul air samples [42]. Even though the concentration of the PAHs was low, the simultaneous exposure to multiple chemicals and high ultrafine particles requires further study. The authors suggest that personal respiratory and skin protection be used appropriately.

#### 3.5.3. Perfluoroalkyl Substances (PFAS)

It has been noted that PFAS may be of particular relevance to firefighting because these compounds are used in turnout gear and are a major ingredient of some firefighting foams [43]. PFAS have been found to play a role in telomere lengthening, subsequently in carcinogenesis (specifically breast cancer), due to cell survival and proliferation [44]. A study reported higher geometric mean concentrations of PFAS found in female firefighters compared with office workers, highlighting the association between firefighting and PFAS exposure [43]. In addition to exposure at work, studies have also shown that firefighters’ exposure in their environment is just as crucial [45,46,47]. According to a recent review, firefighters who were exposed to contaminated drinking water had higher levels of PFAS in their blood compared to those who were considered solely occupationally exposed [45]. Furthermore, a study indicates that firefighters’ serum PFAS levels may be confounded by various factors, such as age, gender, ethnicity, residence in a specific water district, consumption of bottled water, and smoking history [48]. In sum, the existing data on firefighters’ serum PFAS levels are limited, but some exposure to PFAS from working environments, including indoor air and dust [49], is evident.

#### 3.5.4. Polybrominated Diphenyl Ethers (PBDEs)

PBDEs are among the most-used flame retardants and are generally found in domestic materials. Research has shown that firefighters are exposed to higher levels of PBDEs compared to the general population due to gear contamination [50]. While the evidence is limited regarding how PBDEs can lead to carcinogenesis, potential adverse health effects of PBDEs include neurotoxicity, endocrine disruption, and carcinogenicity [51]. Further epidemiological research is warranted to examine the association between firefighters’ PBDE exposure and their prevalent cancers.

#### 3.5.5. Formaldehyde

According to a study by Driscoll et al., various occupations involve exposure to formaldehyde [52]. Firefighting is among these occupations as formaldehyde exposure takes place during fighting fires, fire overhaul and clean-up, and/or back-burning. Since the IARC has classified formaldehyde as carcinogenic to humans [53], the exposure routes for firefighters should be further evaluated so that reactive prevention measures can be better implemented.

### 3.6. Other Causes

Endocrine-disrupting chemicals and their estrogenic and antiestrogenic activity are associated with many diseases, including cancer. Stevenson et al. measured the extracts of used firefighter gear and found that along with phthalate diesters, most extracts exhibit strong antiestrogenic effects, whereas new glove and hood extracts display strong estrogenic activity [54]. Disruption of hormone homeostasis as a result of the exposure may lead to adverse health effects. Probable mechanisms that are involved in cancer formation include transcriptional and epigenetic changes. Transcriptional changes that affect proteins might be associated with inflammation-associated lung disease and cancer as was suggested by a study using mice without airway protection during overhaul [55]. In a study among incumbent and new-recruit non-smoking firefighters, nine microRNAs were identified; six microRNAs with decreased expression in incumbent firefighters are shown to have tumor suppressor activity or are associated with cancer survival, and two of the three microRNAs with increased expression have cancer promotion activities [56].

#### 3.6.1. Lifestyle Factors

In many of the studies we have included in this review, authors often mention a lack of firefighters’ lifestyle information as a limitation when evaluating firefighter cancer incidence and mortality. Obesity and lifestyle factors, such as sedentary behavior, shift work, and alcohol consumption, among others, play a crucial role in disease development, including cancer. Lifestyle plays a crucial role in firefighters’ health and disease prevention. Therefore, lifestyle modifications and interventions promoting dietary intervention should be implemented and serve as effective countermeasures for considerably reducing the risk of various firefighting-associated cancers [57]. Concerning nutrition, Yang and colleagues who assessed 3172 career firefighters’ diet practices reported that most firefighters do not currently follow any specific dietary plan (71%) and feel that they receive insufficient nutrition information (68%); meanwhile, most are interested in learning more about healthy eating (75%) [58]. When presented with written descriptions of diets without names or labels and asked to rank them in order of preference, firefighters most often rated the Mediterranean diet as their favorite and gave it a more favorable distribution of relative rankings (*p* < 0.001). It is worth noting that the Mediterranean diet is associated with significantly decreased cancer risks [59] and a 14% lower risk of overall cancer mortality, an 18% lower risk of colorectal, and a 51% lower risk of head and neck cancer [60]. Therefore, firefighters should engage in healthier eating habits and are recommended to follow the traditional Mediterranean or similar diets that emphasize plant-based foods and healthy fats (e.g., extra-virgin olive oil) to gain protective benefits against cancer. Though firefighters recognize the importance of a healthy diet, following such healthy eating behavior is still a problem. They are often influenced by their colleagues. Hence, counseling and follow-up were often found to be effective [61]. Moreover, cultural barriers and challenges that might be faced when implementing nutrition intervention programs among firefighters should be carefully evaluated. Sotos-Prieto conducted interviews among twelve USA fire service members and five main themes were identified (food environment and food culture; nutrition education; attitudes; perceptions toward making changes; barriers, incentives, and motivating factors to overcome barriers and to improve nutrition) [4]. In particular, the study identified a need for staff and recruits to develop a culture that promotes long-term change, and the participants reported that incentives for good choices and the elimination of certain poor food choices would lead to healthier choices. The study supports an intervention using the education of fire recruits and modifications of the fire academy food environment.

In addition, physical activity among firefighters should be encouraged since it is found that engaging in recommended amounts of activity (7.5–15 MET hours/week) is associated with a statistically significant lower risk of specific cancer types [62]. Moreover, more attention should be paid to improving sleep hygiene and limiting alcohol consumption since, as already stated, both are associated with cancer development [63,64]

#### 3.6.2. Obesity

A US-based study examined the prevalence of overweight and obesity among 677 career and volunteer male firefighters. It demonstrated that almost 76% of career firefighters were overweight or obese (33.5%) compared to 68% of the general US adult population [65]. For this reason, better nutrition and maintaining a healthier weight are important because obesity, apart from the well-known adverse health effects, also increases the risk of many cancers, including esophageal, colorectal, gastric, kidney, thyroid cancer, and multiple myeloma, by as much as 30 to 80% [59,66]. Specifically, a recent IARC report has concluded that there is sufficient evidence to link excess adiposity to thirteen human cancers [66]. Among females, limited evidence suggests that women engaged in heavy physical activity were less likely to be obese contrary to their male counterparts. However, challenges in nutrition intake, inconsistent schedules, time allocation for physical activity, shiftwork, and long-term service result in an increased risk of obesity [67].

#### 3.6.3. Sedentary Behavior

Increased sedentary work due to fire-related calls and high reliance on computer technology is prevalent among firefighters and is believed to result in obesity [68]. Additionally, sedentary behavior seems to increase with the promotion to higher ranks since captains and battalion chiefs spend more time behind a desk than in the field [68]. A study conducted among 8002 black and white adults, evaluating the relationship between sedentary behavior and cancer, showed that replacing sedentary time with physical activity was associated with reduced cancer mortality [69].

#### 3.6.4. Shift Work and Poor Sleep Quality

Firefighters work long shifts including night shifts. Night shift work has shown to be associated with an increased risk of prostate cancer due to the suppression of melatonin synthesis. Melatonin is known to prevent cancer, whereas, with its suppression, there is a disruption of the circadian rhythm and possibly an increase in testosterone levels, potentially leading to prostate cancer development [25]. In addition, poor sleep quality is positively associated with the long-term risk of developing cancer [63].

#### 3.6.5. Alcohol Consumption

When assessing alcohol consumption among 160 career firefighters in the USA, Piazza-Gardner et al. found that firefighters drink more than the general population, including college students [70]. Similarly, Haddock et al., who evaluated alcohol use among 656 male career and volunteer firefighters in the USA, reported high rates of binge drinking among firefighters, specifically 56% among career firefighters, but found that firefighters who drank in moderation had the most positive health and safety attributes compared to those who binge drank [64]. Evidence shows that alcohol is associated with an increased risk of melanoma, prostate, and pancreatic cancers, but decreased risks of kidney cancers and non-Hodgkin lymphoma [64].

#### 3.6.6. Nutrition

Many US firefighters tend to consume comfort and refined/processed foods typical of standard Western diets that are considered unhealthy diets as compared to traditional dietary patterns [71]. Studies have shown that the majority of US career firefighters are overweight, while 33–45% are obese, and less than 30% of career firefighters report following a specific dietary plan [58,72,73]. About 71% of career firefighters in the US do not follow any dietary recommendations [58]. Epidemiological studies observed that firefighters have a high consumption of red meat and fast food [74,75]. Similarly, fat, cholesterol, protein, sugar, and sodium were overconsumed, whereas fruits, vegetables, grains, and dietary fibers were below recommended levels [61,76]. Evidence from surveys shows that the Mediterranean diet could be the most appealing healthy diet option accepted by many firefighters [58], and is consistent with revised US government dietary guidelines [58].

## 4. Discussion

### 4.1. Prevention/Perception

Cancer prevention among firefighters requires a change both in the fire service culture and firefighters’ behavior. The first step for proper prevention methods to be implemented is to evaluate firefighters’ perceptions and concerns on cancer risk associated with their occupation as well as their attitude towards cleaning and carrying out decontamination processes.

In the Jitnarin study, 39 career firefighters and fire service administrations from across the U.S. were interviewed about their perception of cancer and its risk factors [77]. The main themes that emerged were concerns about their activities during the fire and non-fire situations and their lifestyle behaviors. They believed that the agents to which they were exposed throughout suppression and overhaul activities, as well as their contaminated gear and fire trucks’ diesel exhaustion, presented multiple dangers. Additionally, the participating firefighters mentioned that stress, poor diets, lack of sleep, insufficient physical activity, and substance use, such as alcohol and tobacco, could also contribute to cancer development [77]. Anderson et al. collected qualitative data from more than 100 firefighters and found that their perceived cancer risk factors included on-scene exposures, contaminated personal protective equipment (PPE), and exhaust from engines and rescue trucks [78]. Some also believed that cancer might be the result of multiple factors such as genetics, family history, and occupation. Accordingly, the “high risk, high reward” mentality was strongly emphasized in a project carried out by Schaefer et al., which included 57 rookie and experienced firefighters [79]. The participants expressed cancer development as their main concern but also believed that the benefits of having that job outweighed the risks associated with the occupation. Compared to rookie firefighters whose main health concern was just cancer, experienced firefighters reported a concern for chronic illnesses in addition to cancer. In a study by Macy et al., one firefighter expressed his concern about acquiring cancer but said it might be inevitable since they only have one set of turnout gear and they sometimes have to wear dirty or wet gear, endangering themselves and other people at the fire department [80]. Little is known about female firefighters’ concerns and perception of occupation-related cancer risk. When 840 women firefighters from 14 separate countries were allowed to raise any concerns about their health, 54% reported concern about cancer risk, specifically breast cancer [81]. Finally, although alcohol consumption among firefighters has been observed, along with the link between alcohol consumption and pancreatic cancer, a significant increase in the risk of pancreatic cancer among firefighters has not been observed.

In sum, current evidence shows that firefighters have a good understanding of the risks associated with their job. Most of them are aware of occupational exposures as well as lifestyle factors that are potential causes of cancer development.

### 4.2. Attitude towards Cleaning and Decontamination Processes

Harrison and colleagues evaluated 482 USA professional firefighters’ attitudes and found that they generally report positive attitudes; however, they do not engage in frequent and efficient cleaning practices after every fire [82]. Even though showering after each fire seems to be the most regular decontamination measure, 10% of firefighters report rarely or never showering after a fire. In addition, only 15% of firefighters mention routinely cleaning their gear, even though it should be standard procedure. Some barriers to proper decontamination mentioned by the participants were lack of time and concerns over wet gear. Macy et al. examined behaviors related to the retirement, cleaning, and storage of turnout gear [80]. When asked about gear cleaning, 46% reported cleaning as needed, while 19% reported cleaning after every use. Regarding gear storage, over half of all participants reported storing it in a fire department locker, while 28% reported storing it in a private vehicle, and 2% at home. As for gear retirement, 74% mention replacing it less than every ten years, while 26% more than every ten years. According to the National Fire Protection Association, turnout gear should be replaced within ten years of the manufactured date. Cost and lack of funding were strongly emphasized as major barriers to cleaning and retiring turnout gear.

Therefore, for proper intervention methods to be implemented, it is first necessary to change firefighters’ mindset and behavior towards basic decontamination and cleaning practices and educate them on the dangers of the exposures in their occupation. Harrison et al. implemented an intervention across 14 fire stations from 2 fire departments in the USA, which focused on increasing decontamination behaviors after a fire to reduce exposure to carcinogens and consisted of a presentation and videos of firefighter cancer risk [3]. The authors reported significant increases in attitudes, norms, and self-efficacy, decreases in perceived barriers, and increased intention to engage in decontamination processes. While the intervention was highly successful in both fire departments, there were significant differences between the two fire departments in terms of the firefighters’ perceived attitude toward gear cleaning, perceived cancer risk and susceptibility, perceived benefits, and barriers of decontamination, which highlight the need to examine the specific context of the organization in designing interventions. However, no significant differences were found between departments on behavioral intention to clean gear, perceived norms, self-efficacy, perceived cancer severity, and the threat of contamination [3].

### 4.3. Prevention Methods

When designing interventions, it is important to take into consideration issues relevant to the worker, the workplace, and the work arrangement. On a personal level, firefighters should be responsible for cleaning their PPE, showering after fires, performing annual skin cancer screening, applying sunscreen, wearing hats, and avoiding tanning beds. Regarding the workplace, policies on decontamination should be implemented and environmental exposures and skin contaminants should be carefully monitored. As far as work arrangement, some examples include policies to allow fire stations to take some time off for decontamination practices to be implemented, and also providing firefighters with containers to store and transport their gear. These recommendations are highlighted in a study by Caban-Martinez [83] as an example of the Total Worker Health Approach which was developed by the U.S. National Institute of Occupational Safety and Health in June 2011 and is a “comprehensive framework that informs opportunities to sustain and improve worker safety and health through a primary focus on the workplace.” In a study by Louzado-Feliciano and colleagues, it is emphasized that health and safety officers may significantly contribute to the decontamination practices in fire departments, considering that at departments with one or more officers, firefighters follow eight more contamination control practices compared to firefighters at departments with no health and safety officers [84]. It is important that firefighters be more educated regarding occupational dangers and decontamination practices. According to Schaefer et al., experienced firefighters exhibit a positive attitude towards health and safety precautions, which indicates that a change in firefighter culture is currently taking place [79].

### 4.4. Future Directions

Several considerations in future study designs are warranted for more consistent and accurate results on cancer incidence and mortality to be produced. Mainly, larger cohorts, well-characterized exposures, family histories, longer follow-up periods, and more information on lifestyle behavior are necessary. In addition, since most studies include male, Caucasian professional firefighters, more female and racial minority representation should be highly encouraged. This helps to improve the study’s external validity. Additionally, an organizational culture change that favors firefighter health and cancer prevention as well as behavior change on a personal level are both equally crucial for the primary prevention of cancer among the firefighter workforces. Furthermore, existing evidence suggests that lifestyle modifications are crucial factors. Similarly, dietary and other behavioral interventions are effective in preventing firefighters’ adverse health outcomes. Finally, future studies should take into consideration lifestyle information when examining cancer incidence and mortality among firefighters and use other high-risk occupations as a reference to minimize the “healthy worker effect”.

## 5. Conclusions

Firefighting is associated with an increased risk of cancers among career firefighters. Exposures to carcinogens are likely the major risk factor for cancer. Other potential occupational risk factors include shift work, sedentary behavior, and the fire service food culture. Additionally, obesity and lifestyle behaviors such as tobacco use, excessive alcohol consumption, poor diet, and inadequate physical activity have also been associated with an increased risk of cancer.

## Figures and Tables

**Figure 1 cancers-15-02442-f001:**
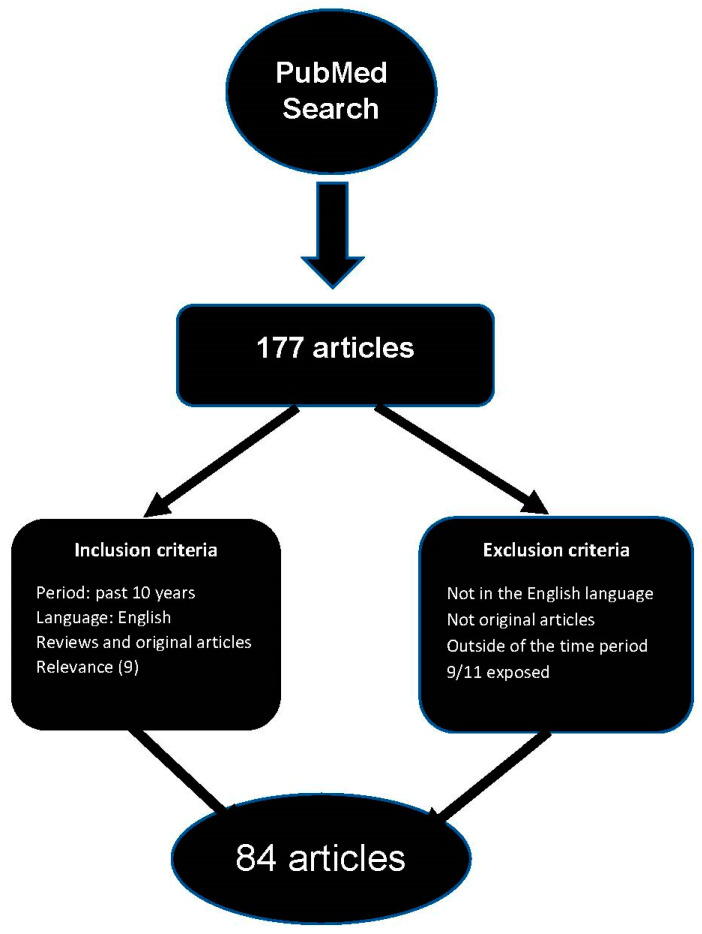
Flow diagram of the search strategy.

**Table 1 cancers-15-02442-t001:** Overall cancer incidence among firefighters.

Author/Year	Cancer Incidence (95% CI)	Study Size	Study Location
North America			
Daniels 2014 [9]	SIR = 1.09 (1.06–1.12) SIR = 1.09 (1.06–1.12) (non-Caucasian male firefighters) SIR = 1.24 (0.89–1.69) (female firefighters)	29,993 career firefighters	USA
Europe			
Bigert 2020 ^a^ [16]	SIR = 1.03 (0.97–1.09)	8136 male firefighters	Sweden
Kullberg 2018 [11]	SIR = 0.81 (0.71–0.91)	1080 firefighters	Sweden
Petersen 2018 [14]	SIR = 1.02 (0.96–1.09)	9061 male Danish firefighters	Denmark
Ide 2014 [15]	Incidence rate/10^5^/year [SD] = 86.5 [64.2] (vs. general population 123.7 [7.9], *p* < 0.001)	~2200 serving firefighters	Scotland
Pukkala 2014 [8]	SIR = 1.06 (1.02–1.11)	16,422 male firefighters	Nordic countries
Oceania			
Glass 2019 [17]	SIR = 1.15 (0.80 to 1.67)	1682 paid female firefighters	Australia
Glass 2016 [10]	SIR = 1.09 (1.03–1.14)	30,057 male firefighters	Australia
Glass 2016 [1]	SIR = 1.85 (1.20–2.73) (High risk of chronic exposure group) SIR = 1.13 (0.80–1.55) (Medium risk of chronic exposure group) ^b^ SIR = 0.40 (0.15–0.87) (Low risk of chronic exposure group)	611 male firefighters	Australia
Asia			
Ahn 2012 [13]	SIR = 0.97 (0.88–1.06)	29,438 firefighters	Korea

SIR = standardized incidence ratio, CI = confidence interval, SD = standard deviation; ^a^ An extension of the Swedish component of the Nordic cohort study (Pukkala et al., 2014 [8]); ^b^ 154 out of 259 subjects in this category were volunteer firefighters.

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
