# Peer review of "Cancer and Potential Prevention with Lifestyle among Career Firefighters: A Narrative Review"

_cancers, 2023, doi:10.3390/cancers15092442_

Round 1

Reviewer 1 Report

I commend the authors for undertaking such a complex, convoluted, yet highly necessary review. While the overall quantity of studies is lacking, the extant scientific literature related to health among firefighters is filled with many inconsistencies (among methodology and findings), which makes systematic reviews difficult and oftentimes "messy". The authors do an adequate job of incorporating studies in a concise, logical fashion. Please see below for more specific thoughts and suggested edits:

1. Line 34-35 needs citation

2. Within Tables 1 & 2, I think it would be helpful to readership to keep the studies organized by continent, but within each continent, list the studies in chronological publishing order. 

3. Line 364 take out "that" after "sufficient evidence"

4. For the use of references 64 and 70, are there parallel studies done in firefighters with these same findings? The strength of your statement would be increased if the findings were among firefighters as opposed to just the general population

5. Line 415 - omit header as this is a continuation of your previous header ("prevention/perception")

6. Line 467- need comma after "self-efficacy"

7. Line 507-509- run-on sentence, needs to be restructured or split into multiple sentences

8. Please confirm there were no studies identified that mentioned pancreatic cancer risk/incidence among firefighters. If so, I think this would be important to note in discussion. Given what we know about alcohol consumption among firefighters, as well as the link between alcohol consumption and pancreatic cancer, I am surprised pancreatic cancer wasn't identified in the studies you reviewed.

9. I am intrigued by the finding noted in line 84 where cancer incidence for females was nearly 1 1/2 times that of males. What do you attribute this to? Especially given that females are generally underrepresented in study samples.

Author Response

Thank you very much for the opportunity to review the manuscript.

Please find the response to the reviewer's comment on a point-by-point basis (below) and a marked copy to see the changes in the main text. All the comments are in the attachment.

Reviewer 1

  1. Line 34-35 needs citation

Citation added (references 1, 2, 3, and 4). Currently at line 37-39.

  1. Within Tables 1 & 2, I think it would be helpful to the readership to keep the studies organized by continent, but within each continent, list the studies in chronological publishing order.

Organized according to the continent and then in chronological order where applicable.

  1. Line 364 take out "that" after "sufficient evidence.

Removed “that” from the sentence. (Currently at line 403)

  1. For the use of references 64 and 70, are there parallel studies done in firefighters with these same findings? The strength of your statement would be increased if the findings were among firefighters as opposed to just the general population.

There are studies which document and describe sedentary behavior/inadequate physical activity among Firefighters, but they have not studied their association with cancer in Firefighters populations (Current references 63 and 69).

  1. Line 415 - omit header as this is a continuation of your previous header ("prevention/perception")

Omitted as suggested. (Now line 454)

  1. Line 467- need comma after "self-efficacy."

Done (Now at line 509)

  1. Line 507-509- run-on sentence, needs to be restructured or split into multiple sentences.

Split into two sentences. (Now line 549-552)

  1. Please confirm there were no studies identified that mentioned pancreatic cancer risk/incidence among firefighters. If so, I think this would be important to note in discussion. Given what we know about alcohol consumption among firefighters, as well as the link between alcohol consumption and pancreatic cancer, I am surprised pancreatic cancer wasn't identified in the studies you reviewed.

A sentence has been added in the discussion section to confirm the findings of our study (lines 479-481).

  1. I am intrigued by the finding noted in line 84 where cancer incidence for females was nearly 1 1/2 times that of males. What do you attribute this to? Especially given that females are generally underrepresented in study samples.

Due to the study’s lack of power and statistical significance, the findings from this study should be interpreted cautiously and may not be generalizable. Hence, additional studies with greater statistical power are needed to provide more robust evidence (lines 122-124)

Reviewer 2 Report

 Comments to Author

Manuscript number cancers-2301958

Title: Cancer and Potential Prevention with Lifestyle among Career Firefighters: A Narrative Review

Overview and General recommendation:

I believe that research is important to field of occupational cancers. This research is narrative review about firefighters workers and risk of cancer. The authors did a collection of papers (systematic reviews and cohort studies) demonstrated that firefighters have statistically significant increases in overall and site-specific cancer incidence and site-specific cancer mortality compared to the general population.

On the one hand, I found the paper to be overall well written. However, it needs some small adjustments to be published.

Comments:

1.         The main is increase of Methods section and to include what kind of studies (only cohorts’ studies) were included in the review? Describe better how the authors selected the studies to include in narrative review.

2.         Would I like to know why studies from other continents were not include? The South America per example did not have studies about it? Describe better what differences have between continents is about socioeconomic conditions.

3.         I would like to see in this paper one Figure or table with a relation of carcinogenic agents and type of cancers in firefighters? Would I like to know the opinion of authors about it?

Author Response

Thank you very much for the opportunity to review the manuscript.

Please find the response to the comment on a point-by-point (below) and a marked copy to see the changes in the main text.

  1. The main is increase of Methods section and to include what kind of studies (only cohorts’ studies) were included in the review? Describe better how the authors selected the studies to include in narrative review.

Selection criteria s primarily based on the search strategy. Line 76-80 and Figure 1.

  1. Would I like to know why studies from other continents were not include? The South America per example did not have studies about it? Describe better what differences have between continents is about socioeconomic conditions.

The underrepresentation of some continents like South America might be due to one of our limitations of literature published only in English (search strategy). As for the sociodemographic conditions, we could present the proportion of developed versus developing countries for each continent. However, a limited number of studies highlighted socioeconomic status for comparison.

  1. I would like to see in this paper one Figure or table with a relation of carcinogenic agents and type of cancers in firefighters? Would I like to know the opinion of authors about it?

The complexity of exposure for firefighters, IARC listed "being a firefighter" as an independent carcinogen instead of breaking down individual exposures. While it might be challenging to create a table or figure listing all possible carcinogenic agents and types of cancers related to firefighters, it would undoubtedly be helpful for readers to understand the cancer risks associated with firefighting.

Reviewer 3 Report

The paper is well-written,  addresses an interesting topic and certainly advances our understanding of cancer epidemiology among carreer firefighters.

However, given the high scientific quality of this review, as well as the accuracy and the thoroughness of the results presented, I do think that the “methods” need to be extensively revised according to the following suggestions:

· since the paper is classified as a “Narrative Review”, the authors should describe their search strategy. For instance, they could, at least, clarify which bibliographic databases they searched for, the search string they used and  if any language limit was applied;

· moreover, inclusion/exclusion criteria, as well as assessment methods applied to each included paper, should be clearly reported;

· finally, it could also be useful to add a flow diagram depicting the different phases of the review (see PRISMA Statement).

Author Response

The paper is well-written, addresses an interesting topic and certainly advances our understanding of cancer epidemiology among career firefighters.

However, given the high scientific quality of this review, as well as the accuracy and the thoroughness of the results presented, I do think that the “methods” need to be extensively revised according to the following suggestions:

since the paper is classified as a “Narrative Review”, the authors should describe their search strategy. For instance, they could, at least, clarify which bibliographic databases they searched for, the search string they used and if any language limit was applied;

moreover, inclusion/exclusion criteria, as well as assessment methods applied to each included paper, should be clearly reported;

Finally, it could also be useful to add a flow diagram depicting the different phases of the review (see PRISMA Statement).

Thank you very much for the opportunity to review the manuscript.

Please find the response to the comment below and a marked copy to see the changes in the main text.

In this narrative review, we did not conduct a systematic search following PRISMA or similar protocols. However, we have revised our manuscript to specify our literature search and the following criteria we used to select studies as described below and in the flow diagram which has been added to the manuscript as Figure 1.

“The search strategy was based on relevant reviews and original publications in PubMed from the past 10 years. Additionally, 9 articles are beyond this time frame selected mainly due to strong evidence. We retrieved a total of 177 articles and included 84 in this review. Inclusion and exclusion criteria are described in the flow diagram (Figure 1).”

Round 2

Reviewer 3 Report

I do believe that the manuscript has been sufficiently improved to warrant publication in Cancers.